# Anti-Wear and Anti-Erosive Properties of Polymers and Their Hybrid Composites: A Critical Review of Findings and Needs

**DOI:** 10.3390/nano12132194

**Published:** 2022-06-26

**Authors:** Zaib Un Nisa, Lee Kean Chuan, Beh Hoe Guan, Saba Ayub, Faiz Ahmad

**Affiliations:** 1Department of Fundamental and Applied Sciences, Universiti Teknologi PETRONAS, Seri Iskandar 32610, Perak, Malaysia; lee.kc@utp.edu.my (L.K.C.); beh.hoeguan@utp.edu.my (B.H.G.); saba_20000009@utp.edu.my (S.A.); 2Department of Mechanical Engineering, Universiti Teknologi PETRONAS, Seri Iskandar 32610, Perak, Malaysia; faizahmad@utp.edu.my

**Keywords:** surface erosion, polymer, nanocomposite, fiber-reinforced composites, thermoplastic polymers, thermoset polymers, surface protection, erosive wear, erosion rate, erosion efficiency

## Abstract

Erosion caused by the repeated impact of particles on the surface of a substance is a common wear method resulting in the gradual and continual loss of affected objects. It is a crucial problem in several modern industries because the surfaces of various products and materials are frequently subjected to destructively erosive situations. Polymers and their hybrid materials are suitable, in powdered form, for use as coatings in several different applications. This review paper aims to provide extensive information on the erosion behaviors of thermoset and thermoplastic neat resin and their hybrid material composites. Specific attention is paid to the influence of the properties of selected materials and to impingement parameters such as the incident angle of the erodent, the impact velocity of the erodent, the nature of the erodent, and the erosion mechanism. The review further extends the information available about the erosion techniques and numerical simulation methods used for wear studies of surfaces. An investigation was carried out to allow researchers to explore the available selection of materials and methods in terms of the conditions and parameters necessary to meet current and future needs and challenges, in technologically advanced industries, relating to the protection of surfaces. During the review, which was conducted on the findings in the literature of the past fifty years, it was noted that the thermoplastic nature of composites is a key component in determining their anti-wear properties; moreover, composites with lower glass transition, higher ductility, and greater crystallinity provide better protection against erosion in advanced surface applications.

## 1. Introduction

Erosion continues to evolve as an important research area; it has attracted the interest of researchers for over 125 years because the reliability of surface integrity is of interest to most industries. Erosion is a major issue since it results in several problems, such as failure/collapse, the degradation of surfaces, severe accidents, and vulnerabilities in many industrial systems and processes. Surface degradation by erosion is a slow but nevertheless continuous and unpreventable process in numerous industries, such as the oil and gas industry; erosion also affects aircraft, steam engines, and the rotor blades of power plant drive turbines, including wet-steam turbines, as well as other turbine plants that operate on wet steam [1,2]. The erosion of surfaces by solid particles is a vigorous process that causes the removal of material from the target surface due to diverse conditions, including viscous fluid flow, the impingement of fast-moving particles, and frictional movement. Indicators of erosion were discussed by Barkoula and Karger-Kocsis [3]. Erosion involves the weakening of components; material deformation; surface cutting, crushing, scratching, and mortification; the absence of directional grooving; abrasion; the detaching of the surface layer of protective coatings; and a decline in the operational life of components, as seen in Figure 1. Due to high maintenance costs and the unplanned nature of stoppages, the control of erosion problems is very time-consuming and expensive. Erosion is seen as a major challenge in the industries it affects, where its impacts on plants include:Halting activities;Shortening productive service life;Decreasing performance;Requiring material replacement, including repair and maintenance costs;Decreasing productivity and efficiency;Reducing revenue;Impacting safety (causing explosions, fires, and discharges of toxic products);Health impacts (personal injuries and the contamination of the environment due to the escape of toxic products).

Developments in production methods and advancements in materials and coatings that offer exceptional performance and distinct properties represent one of the most crucial imperatives in modern industrial techniques [4,5]. During their lifetime, surfaces are subjected to many hazardous influences that cause their deterioration, such as thermal and chemical degradation, stress corrosion, external loads, abrasive particles, and pressure [6,7]. Understanding the kind of activities that take place when a surface comes in contact with gases, chemicals, solid hard particles, or viscous fluids can allow us to predict the lifespan of affected surfaces and utilize the finest materials for the relevant application and working conditions [8,9].

This study aims to explore the techniques utilized for wear testing, the factors leading to surface erosion, and the wear behavior of neat polymers utilized in the past fifty years; to correlate modifications in erosive properties, the available data for hybrid glass and carbon-fiber-reinforced composites are compiled from the literature for the past twenty-five years. The study further proposes to identify gaps in the literature to aid researchers in directing their future work, with the goal of effectively utilizing optimal materials in industries where surfaces are exposed to wear-prone environments.

### 1.1. Techniques Utilized for Wear Testing

Several methods of evaluating the wear resistance of materials are available [10,11]. Subject to tribological system impacts, all testing methods can be divided into two categories:Contact methods that involve the testing surface material in common contact with another material, including or not including supplementary media.Methods utilizing a flow of coarse erodents or loose solid coarse particles in a liquid or gas.

In the first class of techniques, erosion is determined after a specified length of time sliding the surface to be tested against the surface of a reference material. In the second group of procedures, loose abrasive particles are dispersed onto the analyzed surface at a specified angle using compressed air or water as a medium. Considering both of the above cases, the most extensively utilized methods are:Air jet erosion testers;Slurry jet erosion testers;Water droplet erosion testers;Dry erosion testing with loose particles of the erodent;Wear testing using solid particle accelerators;The Taber method;Silica blasting;Shot blasting.

### 1.2. Factors Affecting Erosion

Erosion by fluid or particles is a vigorous phenomenon that includes gradual harm to a surface caused by an impinging material because of repeated collisions and interactions. Based on the physicomechanical properties of the substrate surface or the substrate coating, the wear of the surface also differs according to the flow of the abrasive material [12,13]. Numerous authors have reported the outcomes of several assessment parameters and described the effect of factors such as a material’s nature, form, and amount of fiber on the performance of various polymers under erosive environments. In addition, other important parameters include the angle of attack between the velocity vector of the erodent and the impinging surface; the erodent velocity; the size of the erodent and its form; the physicomechanical characteristics of the impinging material (hardness and impact toughness); and the flow temperature, humidity, and pressure. Some of the important factors related to erosion are summarized in Figure 2.

There have also been numerous studies carried out to explore the prompting contributors to solid particle erosion, comprising impact velocity [14,15], impingement angle [16], particle flux rate [17,18], temperature [19], char of the impinging surface, and parameters such as the shape of the erodent, its dimension, and the toughness of the materials. The complicated nature of the solid particle process is exposed by these variables. Even though wear phenomena caused by solid particle impacts have been extensively studied, there are still several aspects remaining to be explained that may be explored by researchers, including the production of innovative materials to decrease erosion and the study of an inclusive erosive resistance mechanism [20].

Mathematical theoretical models using artificial neutral network approaches and linear regression have also been successfully applied using factorial design. It has been suggested that the extent of the exposure to erosive conditions, the inclination angle, the velocity of erodent, the radial distance, the path of the sample’s rotation in the medium, and the distance tracked all have an influence on the wear resistance of materials [21,22].

Overall, an ideal erosion-resistant material should have an elevated surface area/volume ratio, high hardness, high adhesive strength, small filler size, and a good dispersion of filler; these characteristics result in better mechanical properties, namely scratch resistance and high ductility without strength loss, as presented in Figure 2. To achieve these parameters, synthesizing nanocomposites utilizing erosion-resistant polymers, i.e., thermoplastic/ductile polymers, is the finest choice. Nanocomposites as a material have integrated nano-sized fragments into their matrix; as a result, when utilizing ductile thermoplastic polymers, it is observed that they significantly improve the properties related to erosion resistance, such as strength, modulus, hardness, toughness, and dimensional stability [3,23,24,25,26].

For specific mechanical and physical applications, the wear resistance of the base material must be identified. The vital parameters in the wear process and their role in various classes of composites and polymers have been highlighted by Barkoula and Karger-Kocsis [3]. The role of filler and/or fiber has been discussed in detail in the literature, and unpredictable results and the complexity of the process are seen in studies on erosion [27]. It has also been revealed that homogeneous mixing and diffusion of nanoparticles, the type and content of the polymer matrix, and the arrangement of the fiber used for the reinforcement of the polymer substrate are the important parameters that control the performance enhancement of nanocomposites. The shape, nature, and amount of fiber are critical parameters for the fabrication of composites and govern their thermal, mechanical, and thermomechanical behavior. Generally, the main factors determining the distribution of the filler are the surface energy of the elements, the matrix–filler compatibility, and the processing methods. The adhesion between surface and matrix is also critical, and it is dependent on the mechanical adherences between the filler matrix and the matrix surface.

Based on a study by Zhang et al. [28], it was concluded that the thickness of the layer of the surface film used as protection changes the wear resistance. For a highly thermally conductive substrate, the study of the protective coating at room temperature proved that erosion declines with growing film thickness. However, for very large film thicknesses, increasing film thickness results in an increase in erosion under certain thermal parameters.

To study erosion resistance and relate it to several mechanical properties of neat materials and their hybrid composites, researchers have carried out effective tests [29]. Deformation brought on by erodents is linked with elevated strain rates of approximately 10^5^–10^6^ s^−1^; therefore, predicting stress conditions is difficult and correlating mechanical properties with erosion rates is also very complicated. Similarly, it is also tough to foresee their comparative impacts because there is a series of certain mechanisms that encourage the process of erosion. This is concluded based on a literature survey that found all these results in physical effects such as filler breakage/damage, fragmentation, pull-out, denaturing, debonding, etc. In addition, the reliance of the erosion rate of polymer composites on experimental factors is dependent on the wear process [27].

### 1.3. Erosion Variables

The properties of a material play a major role in the erosion mechanism and the erosion rate in wear environments. Hutchings et al. [30] proposed that the severity of wear can be determined using a dimensionless wear coefficient/erosion efficiency (η), and the below equation can be utilized to calculate it:η = 2EH/ ρv^2^
where E = the erosion rate at a steady state, ρ = the density of the protective surface that is eroded, v = the particle’s velocity of impingement, and H = the hardness of the impinging surface.

The erosion efficiency varies as a function of hardness; as the material becomes harder, the removed segment of the crater volume is increased. For the wear of neat resins, the value of k typically ranges between 10^−3^ and 10^−4^. The mechanism and nature of erosion can be predicted to some extent using the wear coefficient/erosion efficiency (η). It takes into consideration that a dislodgment of surface material from the crater without any damage results in micro plowing (i.e., no erosion will take place when η = 0). In contrast, if the surface erodes by ideal micro-cutting, then η = 1.0 or 100%. It has been observed that a very low value of η (η ≤ 100%) results in surface loss by platelet formation or cutting in the form of a lip. Such material results in surface losses or fractures by continuous impacts and is said to be ductile material. Brittle materials usually erode by spalling and the interlinking of lateral or radial cracks, and the erosion of their surface results in material removal in the form of large chunks. In such cases, quite a high value of η value is observed, i.e., a value greater than 100% (η ≥ 100%).

The velocity of impinging particles is used to calculate the erosion rate. The impact velocity (v) can be correlated to the erosion rate (E) using the expression:E = Kv^n^
where n = the exponent of velocity and K = the proportionality constant.

At various impact angles, the magnitudes of K and n can be obtained using this power equation by the least-square fitting of data points. The velocity exponents are found to range from 1.4 to 3.0 for several polymers at different impact angles. Mostly, greater values of n are associated with sharper impact angles for polymers. Pool et al. [30] concluded that the value of the velocity exponent n ranges between 2 and 3 for polymeric materials acting in a ductile manner, whereas the polymer hybrid composites show a brittle trend, and their magnitudes of n vary from 3 to 5 [31].

## 2. Brief Details on the Historical Utilization of Anti-Erosion/Anti-Wear Materials

Erosion by solid particles results in the thinning of components, surface crumpling, abrading, roughening, scratching, deprivation, and a drop in the practical lifespan of the affected components. Hence, wear resulting from particle impact has been deemed a critical challenge; it is responsible for many breakdowns, collapses, and failures in engineering applications. A robust struggle is ongoing to develop coatings that impede the activity of metal erosion, which is estimated to cost U.S. industries more than two hundred billion dollars yearly [32,33]. A summary of the historical utilization of different materials for this purpose is presented in Figure 3. Earlier understandings of the erosion process proposed that materials with greater hardness, for example, metal and ceramic-based materials, could deliver exceptional wear resistance. As such, conventionally, single-phase coatings of pure metals, metal nitrides, metal carbides, and alloys have been employed [34,35,36]. However, these coatings were lacking in terms of erosion-resistant performance and did not perform well for normal angles due to their high internal stress. In addition, upon impact with sand particles, some metals may spark [24].

To overcome problems with metallic coatings, the second-generation erosion-resistant coatings were introduced: multi-layered structures constituting metal/ceramic materials. These materials resulted in new drawbacks, sometimes in the form of cracks. Consequently, coatings comprising polymers (polymeric coatings) were introduced, owing to characteristics inherent to the polymers [37,38].

Several theories relating to tribology and polymer tribology have been developed [13,39,40]. Many researchers have explored the erosion process, and various analyses of erosion phenomena have been proposed that could explain the activation of the process and the degradation of metallic surfaces/tubing/piping, as seen in Figure 2. Several models and tests have been developed to aid in understanding the erosion process, and modifications to products have been discovered in some cases. There are both limited and complex theoretical and experimental models on polymer composites. The majority of them explain scratch wear as a result of strikes by solid erodents that are transferred by air or by a variety of liquids present in the working environment.

The literature includes research whose authors discovered that plastic polymer-based materials displayed characteristics such as elevated deformation, plasticity, malleability, and great flexibility. It was assumed that they had the ability to absorb and release impact energy very productively in a cyclic way during the deformation and restoration process. Therefore, the employment of polymer-based materials was attempted, and they effectively exhibited better wear performance in erosive situations than ceramic-based materials or metal. Hence, the routinely used metal and metal/ceramic-based anti-erosive protective composites were gradually replaced by polymer materials. Polymeric materials can be handled easily and have better processability, low cost, good erosion resistance, and a greater ability to absorb impacts. Detailed information about the types of polymers and their composites is summarized and presented in Section 3.

### Historical Overview of Simulation Techniques for Erosion Studies

Harsha et al. [41] conducted experiments on polyether ketone (PEK) and its composite under distinct experimental parameters and predicted their erosion rates using artificial neutral networks (ANNs). Arul et al. performed research [42] that explored the solid particle erosive wear behavior of a hybrid composite of polyester and multi-components comprising polyester, granite particulates, and glass fibers. A mathematical model was developed using the law of conservation of energy, and validation was carried out through a planned set of experiments. For this series of experiments, samples were tested in an air jet erosion tester using Taguchi’s orthogonal array, and it was concluded that impact angle, impact velocity, erodent nature, erodent dimensions, and filler content were the critical elements for predicting erosion rate. Taguchi’s approach facilitates the estimation of ideal conditions that can deliver an insignificant wear rate. Wee et al. [43], focusing on the underlying safety hazards and financial reliability of the petroleum industry, investigated the sand erosion behavior of pipelines by employing computational fluid dynamics (CFD).

Zhang et al. [44] used finite element (FE) simulation to elucidate the role of thickness on particle erosion. Seward et al. [45] proposed a procedure for generating a dynamic FE friction model using open-loop tribological testing. The effects of fiber angle, slip rate, and contact pressure for machining simulations were studied, and a relationship was established between tribology, machining, and the FE simulation by applying a cutting-edge tribological study. Zhang et al. [46] demonstrated the manufacture of low-cost, robust, long-lasting, and multifunctional materials for erosion resistance and ice release for shielding wind turbine blades and towers. It was proposed that the novel materials could be utilized to improve the steadfastness of power generation in icing conditions.

Ibrahim et al. [47] presented the Taguchi–Deng approach to optimize tribological factors such as weight, grit dimension, distance, and velocity; they predicted the tribological performances of polytetrafluoroethylene (PTFE) composites modified with carbon and bronze by applying the pin-on-disk configuration.

The above techniques described in the literature predict the roles of different parameters and have been found useful and relevant to wear applications.

## 3. Research on Erosion Resistance Using Polymers

It has been shown in the literature that polymers and their associated composites are extensively utilized materials in erosive wear situations [3]. Since the beginning of the 1980s, research work focusing on the erosion behavior of materials has expanded from metals to fiber-reinforced polymer materials [30,48,49,50]. To tackle the problem of erosion, the polymeric coatings were modified with fillers; the important considerations for the selection of fillers are the properties of the filler, the particle size distribution, the ease of dispersion, good suspension in the coating solution, and rheological effects. The modification of composites by suitable fillers has resulted in the enhancement of mechanical and erosive characteristics; in some cases, it has also incorporated other distinctive characteristics, such as magnetic qualities and conductance to electricity or heat [51,52,53]. Some research works have reported on the use of hard ceramics to achieve surfaces with individual improvements in erosion resistance [54,55]. In order to absorb surplus energy during erodent impacts, there have also been efforts to fabricate laminated composites modified with both multidirectional carbon fibers and glass fibers, but the enhancements in wear performance were observed to be insufficient [56,57]. For the manufacture of wind turbine blades, pipings, and automobile frames, the use of characteristic anti-erosion polymer-based composites has been reported [20,58]. The utilization of polymer-based materials is growing to include aerospace applications, windmill blades, and automobiles. It is well known that neat polymers exhibit a higher erosion rate in comparison to polymer composites [59,60,61].

### 3.1. Thermoset and Thermoplastic Polymers

As discussed earlier, the particle erosion behavior of polymers has been explored by many researchers. The specific materials studied include polystyrene [62], polypropylene [63,64], nylon [65], polyethylene [66], ultra-high-molecular-weight polyethylene [67], polyetheretherketone [68], polycarbonate and polymethylmethacrylate [69], epoxy [70], bismaleimide [71], elastomeric materials [72,73], and polymer composites. Several thermoplastic/thermoset-based hybrid materials have been studied for erosive performance [74,75]. The details of the minimum reported erosion efficiency and erosion rate under optimized conditions are presented in Figure 4 and Figure 5, respectively. Owing to the difficulty of visualizing such a wide range of values in a presentable form, multipliers for some polymers are given in the labels. The literature has shown that the use of ductile elastomeric polymers is a popular choice for erosion resistance. Thermoplastic composites have competed with epoxy composites due to their elevated strength and stiffness, high strain towards malfunction, low water absorption, and infinite shelf life. Consequently, these composites are frequently utilized in parts where wear problems are of major concern.

The available data for neat thermoset and thermoplastic polymers are presented in Table 1 to demonstrate the effects of the matrix. It is obvious that the key factors for correlating the effectiveness of materials with erosive wear are the density, hardness, and ductility of the matrix. Out of the two classes of polymers, thermosets have shown a greater tendency to erode than thermoplastics. In thermoplastic polymers using silica as the erodent, PEK, PA6, PA66, and UHMWPE have shown minimal erosion rates. However, erosion resistance is not solely an inherent material property; several parameters, such as the type, nature, and velocity of the erodent, also affect the utilization of the matrix towards its end application [30,31]. The erosion resistance of polymers has been studied with different erodents, such as silica, glass beads, quartz, corundum, and silica carbide. In oil pipes, the erodent is usually silica; when it is used as the erodent, the lowest reported erosion rate is that of polyamide 6 due to its superior mechanical properties [65]. The erosion of thermoset polymer has been reported using corundum and quartz [3,70]. Few works are available that study the erosion of thermosets. Epoxy and polyurethane have been shown to be brittle, have less malleability, and possess higher glass transition values [30].

For the prediction of the erosion of polymers, the thermoplastic or thermoset nature of the resin is the deciding factor. Generally, thermosetting polymers demonstrate brittle behavior, while thermoplastics exhibit ductile behavior. Erosion efficiency values for thermoplastics using a heating rate of 5 °C/min for 2 h under an inert atmosphere of 3% are less than carbonization temperature at 400 °C and around 100% for thermosetting polymers [15]. A literature survey revealed that polyamide 6 (PA6), polyphenylene sulfide (PPS), and polyethersulfone (PES) are among the thermoplastic polymers that show minimal erosion in their neat forms [24]. So, despite the dynamic nature of the erosion process, thermoplastics are a good material of choice for resistance against material loss in erosive wear situations. It has been reported that thermoplastic composites are better than thermoset epoxy and even aluminum–copper and aluminum–lithium alloys in the aircraft industry [76]. This behavior can be explained by their high strength and stiffness, greater strain to failure, unlimited shelf life, and minimal water absorption in comparison to conventional epoxy composites [77].

### 3.2. Polymer Composites

Besides simple polymers, fiber-reinforced polymers are also well regarded because these composites deliver additional advantages, including their outstanding mechanical properties, freedom in fabrication, strength, lightweight character, and tailorable anisotropy for specific applications [78]. The fiber composite materials are applied to reduce the wear rate and support various applications from daily-use appliances to high-tech manufacturing and aircraft structures. The composites explored using glass fiber and carbon fiber reinforcement are summarized in Table 2. It was shown that there was an unpredictable trend in the increase or decrease in the erosion rate with the addition of glass or carbon fiber. For instance, PEK in neat form has a lower erosion rate than its 10% GF modified composite. If the amount is increased to 20% or 30% GF, the rate of erosion decreases further as compared to the neat polymer under the same impingement conditions. So, the relative percentages and experimental conditions determine the ultimate loss of surface material by erosion.

A serious concern also exists in the case of polymer composites: in some works, relatively low erosion efficiencies have been observed compared to metallic substrates when fibers were used as reinforcement [79,80]. Researchers worked to fabricate polymer laminated composites using multidirectional fibers as reinforcement to absorb additional impact energy from impinging solid particles, but the enhancements in erosive performance were found to be insufficient in the case of both carbon fibers and glass fibers [56,57]. Mostly, polymer composites contain two or more supporting phases bearing a single matrix. The supporting phase may be fibers or filler granules/particles in a mixture with fibers or filler granules/particles, etc. Such phase blends may deliver interactive supporting effects resulting in enhancements to the mechanical properties of the composites [81].

The cost of the manufacturing process is usually reduced by manufacturing polymer composites with a blend of the finest tribological, mechanical, and thermal characteristics. In the last few decades, hybrid composite materials have been productively deployed in erosion wear environments.

In the field of nanocomposites, there has been diverse research on using epoxy and polyurethane resin-based nanocomposites to synthesize coating materials because of their advantageous mechanical properties, such as excellent adhesion strength, high modulus, and low creep; however, the resulting materials sometimes prove brittle in different combinations [82,83]. Different percentages of nanoparticles are used as fillers to overcome the brittle behavior of composites. Nanoparticles, such as silica (SiO_2_), can increase fracture toughness. The use of polymer nanocomposites has attracted considerable attention as a method of enhancing polymer properties and extending their utility by using molecular or nanoscale reinforcements rather than conventional particulate-filled composites [84,85]. Composites that utilize nanofiller reinforcement are light in weight and biodegradable, and they have higher thermal stability, improved mechanical strength, and substantial improvements in rheological properties, making new desired material applications possible [85,86,87,88].

Coating technology is now actively using nanomaterials in different industrial sectors owing to their exceptional capabilities. The advantages include surface toughness, hardness, adhesive strength, temperature resistance, erosion resistance, corrosion resistance [89], the improvement of tribological characteristics, etc. The ease of handling nanocoatings allows for thinner and smoother applications and flexibility in the design of internal parts; this has proved beneficial for improving efficiency and fuel economy and lowering maintenance costs. In the literature, wear studies of composites include a variety of composites with different applications; most of the data presented concern their mechanical and thermal properties, and comprehensive erosion parameters are not discussed.

Patnaik et al. [90] reported that a polyester hybrid composite modified with glass fiber and alumina demonstrated a greater improvement in erosion performance than neat polyester composites. Hamd et al. [91] reviewed the effectiveness of the polymer matrix. The relevant data are presented in Table 3. Based on the data, it is assumed that information on mechanical properties, scratch tests, and thermal properties can be utilized to predict erosion phenomena depending upon the application.

**Figure 5 nanomaterials-12-02194-f005:**
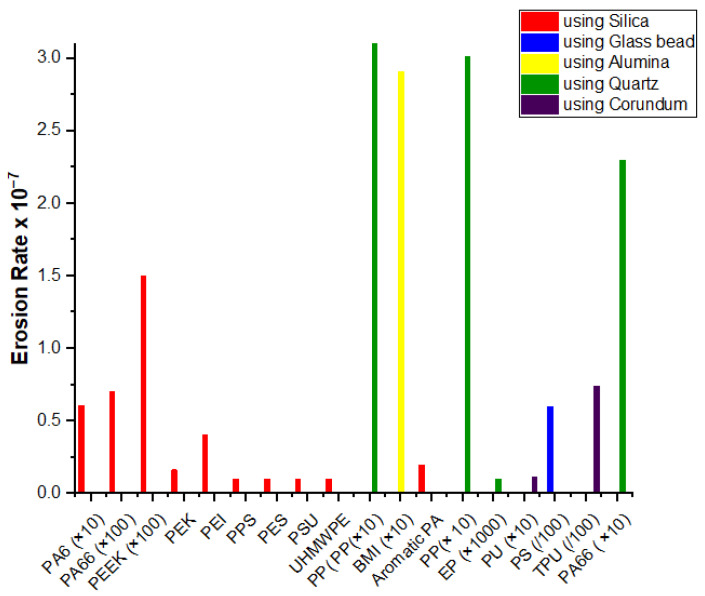
Minimum reported erosion rate (×10^−7^) of neat polymers at optimized impact velocity using different erodents [92].

Korostelyov et al. [2] explored the use of simulation techniques to predict the efficiency increases of soldering wet-steam turbine rotor blades and applying materials, coatings, inserts, and layers. The mathematical experiments used to predict the wear resistance of materials used in wet-steam turbine blades were explained.

Pool et al. [28] highlighted that polyurethane, epoxy, or EP/polyurethane composite coatings that comprise nanofiller reinforcements, such as zinc oxide, alumina, titanium oxide, silica, clay, and nanocarbon or other carbon-built materials can be utilized for offshore pipelines that are susceptible to erosive environmental conditions. It was revealed that the deterioration process could be controlled using the above surface coatings, and such coatings are renowned in terms of production reassurance, ease of use, and pertinency. To tailor the matrix properties of epoxy, which is an inflexible and brittle polymer, polyurethane is a potential candidate for mixing due to its favorable chemical and mechanical properties. It was concluded that synthesizing hybrids of these polymers leads to improvements in characteristics such as mechanical performance; improvements in the temperature of the glass transition and the flexibility of processing can be achieved depending upon the nanofiller.

Chellaganesh et al. [92] studied one of the main issues in modern research on solid particle air jet erosion. In this work, an epoxy-based Kevlar–hemp (40:60 wt.%) hybrid composite material was studied with and without filler. The research was carried out in eight proportions following from 2 k (where k = 3) factorial design.

**Table 1 nanomaterials-12-02194-t001:** Data on the erosion rates of neat thermoset and thermoplastic polymers.

Sr. no	Nomenclature	Material Density (g/cm^3^)	Hardness Value	Erodent Used	Impingement Velocity (m/s)	Erosion Rate (m^3^/kg)	Coefficient of Wear	Remarks	Year	Ref
**Thermoplastic Polymers**										
1.	PA66	1.14	11.1	Quartz	243	2.30 × 10^−6^	N.R.	Study of material response in erosive situations in relation to matrix–particle interaction	1970	[70]
2.	Polystyrene	1.16	11.9	Glass beads	15	0.06 × 10^−9^	N.R.	Study of erosion behavior of PS	1981	[62]
					20	0.14 × 10^−9^				
					40	0.16 × 10^−9^				
3.	PA6	1.15	10.2	SiO_2_	80	9.08 × 10^−9^	N.R.	Study of erosive wear of PA	2001	[65]
4.	PA66	1.14	11.1	-do-	80	7.02 × 10^−9^				
5.	Aromatic PA	1.12	11.7	-do-	80	19.64 × 10^−9^				
6.	TPU-1	1.14	14.5	Corundum	70	0.74 × 10^−9^	N.R.	Study of erosion process and factors affecting it	2002	[3]
7.	PEK	1.32	32.6	SiO_2_	39	4.92 × 10^−9^	N.R.	Study of morphology and possible wear mechanism	2003	[61]
8.	PEI	1.27	40	SiO_2_	25	4 × 10^−8^	1.26 × 10^−2^	Relation of erosion rate and mechanical properties of the neat polymer was studied	2008	[24]
9.	PEEK	1.30	28			3 × 10^−8^	6.06 × 10^−3^			
10.	PEK	1.30	34.4			5 × 10^−8^	6.88 × 10^−3^			
11.	PPS	1.40	26.5			2 × 10^−8^	4.46 × 10^−3^			
12.	PES	1.37	24.2			3 × 10^−8^	5.12 × 10^−3^			
13.	PSU	1.24	21.4			1 × 10^−8^	4.76 × 10^−3^			
14.	UHMWPE	0.93	N.R.			4 × 10^−9^	N.R.			
15.	PEEK	1.3	32.6	SiC	34	0.25 × 10^−8^	0.25 × 10 ^−8^	Study of mechanism and wear process	2017	[93]
**Thermoset Polymers**										
16.	PP	0.91	5.40	Quartz	243	3.10 × 10^−6^	N.R.	Details of material interaction with the particle in erosive conditions	1970	[70]
17.	EP	1.2	40.8			10.0 × 10^−5^				
18.	Bismaleimide	1.33	54.1	Alumina	60	3.44 × 10^−6^	96	Study of erosion behavior	1991	[71]
19.	PU-1	1.26	18.1	Corundum	70	1.11 × 10^−9^	N.R.	Study of erosion process and factors affecting it	2002	[3]

Abbreviations used: PA = polyamide, PP = polypropylene, EP = epoxy, PU = polyurethane, TPU = thermoplastic polyurethane, PEK = polyether ketone, PEEK = polyether ether ketone, PEI = polyetherimide, PPS = polyphenylene sulfide, PES = polyether sulfone, PSU = polysulfone, UHMWPE = ultra-high-density polyethylene, N.R. = not reported.

**Table 2 nanomaterials-12-02194-t002:** Data on the erosion rates of GF/CF-reinforced composites.

Sr. no	Nomenclature	Material Density (g/cm^3^)	Hardness Value	Erodent Used	Impingement Velocity (m/s)	Erosion Rate (m^3^/kg)	Coefficient of Wear	Remarks	Year	Ref
1.	BMI + 20 bisphenol	1.29	53.4	Alumina	60	6.14 × 10^−6^	193	Study of erosion behavior	1991	[71]
2.	BMI + 40 bisphenol	1.23	34.9		60	3.31 × 10^−6^	63			
3.	BMI + 60 bisphenol	1.29	53.4		60	3.44 × 10^−6^	74			
4.	EP unidirectional + CF 56%	1.51	40.7	Steel balls	45	0.88 × 10^−7^	N.R.	Response in solid particle erosion conditions of unidirectional CF- and GF-reinforced epoxy composites	2003	[94]
5.	EP unidirectional + GF 56%	1.88	63.7		45	1.38 × 10^−7^				
6.	PEK	1.32	32.6	SiO_2_	39	4.92 × 10^−9^	N.R.	Study of morphology and possible wear mechanism	2003	[61]
7.					68	1.62 × 10^−8^				
8.					90	4.22 × 10^−8^				
9.	PEK + 10% GF	1.38	35.5	SiO_2_	39	6.23 × 10^−9^				
10.	PEK + 20% GF	1.44	39.5		39	0.83 × 10^−8^				
11.	PEK + 30% GF	1.53	43.7		39	1.04 × 10^−8^				
12.	PEEK	1.31	333.5		39	1.05 × 10^−8^				
13.	PEEK + 30% CF	1.36	50.6		39	1.82 × 10^−8^				
14.	PEI	1.27	41.9	SiO_2_	30	6.69 × 10^−9^	N.R.	Mechanical properties and possible wear mechanisms discussed	2007	[95]
15.	PEI + 20% GF	1.42	42.1		30	5.98 × 10^−9^				
16.	PEI + 30% GF	1.51	46.7		30	5.63 × 10^−9^				
17.	PEI + 25% CF	1.7	41.7		30	0.75 × 10^−8^				
					52	1.25 × 10^−8^				
					60	5.50 × 10^−8^				
					88	9.51 × 10^−8^				
18.	PEEK	1.30	28	SiO_2_	25	3.0 × 10^−8^	6.06 × 10^−3^	Mechanical properties and possible wear mechanisms discussed	2009	[96]
19.	CF/PEEK	1.56	60			3 × 10^−9^	3.12 × 10^−2^			
20.	GF/PEEK	1.99	101			3 × 10^−9^	7.07 × 10^−2^			
21.	CF/PEKK	1.58	85			1 × 10^−9^	5.30 × 10^−2^			
22.	GF/PEKK	2.08	112			2 × 10^−8^	8.45 × 10^−2^			
23.	PPS + 20% GF	1.48	35.8	Silica sand	N.R.	3 × 10^−8^	N.R.	Study of erosion rates	2009	[97]
24.	PPS + 30% GF	1.6	37.5			3 × 10^−8^				
25.	PPS + 40% GF	1.6	110			2 × 10^−8^				
26.	PEEK	1.3	32.6	SiC	34	0.25 × 10^−8^	N.R.	Study of mechanism and wear process	2017	[93]
27.	PEEK + 30% GF	1.51	40.9			0.40 × 10^−8^				
28.	PEEK + 30% CF	1.38	37.7			0.50 × 10^−8^				

Abbreviations used: EP = epoxy, PEK = polyether ketone, GF = glass fiber, CF = carbon fiber, PEEK = polyether ether ketone, PEI = polyetherimide, PPS = polyphenylene sulfide, BMI = bismaleimide, N.R. = not reported.

**Table 3 nanomaterials-12-02194-t003:** Reported composites with properties related to erosion.

Sr. no	Nomenclature	Material Density (g/cm^3^)	Hardness Value	Erodent Used	Impingement Velocity (m/s)	Erosion Rate (m^3^/kg)	Coefficient of Wear	Remarks	Year	Ref
APS modified TiO_2_ nanoparticles/in PU										
1.	UT-1 (PU with 1% neat nano titania)	N.R.	167	N.R.				Only Tg was reported	2009	[83]
2.	TT-1 (PU with 1% modified nano titania)	N.R.	185	N.R.				Only Tg was reported	2009	[83]
3	TT-2 (PU with 2% modified nano titania)	N.R.	187	N.R.						
4.	TT-3 (PU with 3% modified nano titania)	N.R.	187	N.R.						
5.	CNT/epoxy composites	N.R.						Only graphical data for the wear study is available	2014	[98]
6.	Epoxy neat resin	N.R.	0.17	N.R.				Study of mechanical and anticorrosive properties of surface	2015	[99]
7.	Resin + 0.1% graphene		0.19	N.R.						
8.	Resin + 0.4% graphene		0.41	N.R.						
9.	Resin + 0.7% graphene		0.51	N.R.						
10.	Graphene Oxide-reinforced silicone-acrylate resin	Study of mechanical stability to be used as a coating for erosion resistance in mechanical corrosion coupling environments							2015	[100]
11.	Epoxy/clay nanocomposite with organically modified montmorillonite	Study of mechanical strength, electrical conductivity, flammability, and thermal stability for applications in aerospace, adhesives, and coating industries							2016	[101]
	Silica-filled epoxy nanocomposite							Effect of pyrogenic silica on epoxy resin for use as anti-cavitation painting	2017	[82]
12.	H (Neat)	1.11	75	SiO_2_	N.R.					
	H_3_SiO_2_ (3 wt%)	1.12	61							
	H_5_SiO_2_ (5 wt%)	1.13	60							
13.	Epoxy/graphene-reinforced composites	Composites were qualitatively and quantitatively evaluated in terms of contact angle, surface roughness, adhesion to the substrate, corrosion resistance, and abrasion resistance for oil and gas pipelines							2018	[102]
14.	Ni-P-nano-NiTi composites	Only scratch test and indentation test were performed							2019	[103]
15.	Mechanical characterization of wood apple- and coconut shell-reinforced hybrid composites	Fabrication of coconut and wood apple shell powder-reinforced epoxy composites and experimental investigation of mechanical properties							2020	[104]
16.	Characterization of carbon fiber-reinforced polyphenylene sulfide composites prepared with compatibilizers	Interfacial adhesion was studied using SEM and DMA (dynamic mechanical analysis) with the addition of Joncryl between carbon fiber and polyphenylene sulfide							2020	[105]

Abbreviations used: H = epoxy, PU = polyurethane, APS = aminopropyl trimethoxy silane, N.R. = not reported.

The erodent volume, air jet velocity, and angle of impingement were evaluated as process parameters. In addition, an analysis of the surface morphology was conducted using an electron microscope. Wear rate was calculated based on mass change relative to time. It was demonstrated that filler improved the wear properties of the composite material compared to resin.

Maclean et al. [103] studied oil and gas pipelines under drastic chemical and mechanical conditions that could lead to devastating failure. Nickel–phosphorous (Ni–P) electroless coatings are an exceptional commercial candidate for protection against the corrosion and wear of oil and gas pipelines. However, its demonstrated inferior toughness has reduced its use in products where superior dent and scratch resistance are required. To improve the toughness and scratch-resistant properties of the Ni–P coating, the authors integrated nano-sized particles of a super-elastic NiTi alloy into the coating to obtain a composite. Scratch tests under increasing and constant loads were performed to determine the effect of the NiTi nanoparticles on wear damage. In addition, indentation tests were performed to evaluate the crack methods and dent resistance of the composite coating. It was concluded that the addition of super-elastic nanofiller particles into the Ni–P matrix resulted in transformation, microcracking, deflection, toughening, and crack bridging but still delivered a significant improvement of the substrate.

Abenojar et al. [82] evaluated the influence of pyrogenic silica on epoxy resins by using different loads of nano-silica in SiO_2_/epoxy nanocomposites. The study discovered that adding nano-silica had a considerable effect on the curing reaction and cavitation erosion, as well as on the wear, mechanical, and thermal properties. Overall, a plasticizing effect was noted with the addition of nano-silica. It was concluded that a decrement in cumulative mass loss and an improvement in cavitation erosion, in terms of collective erosion and erosion rate, were observed for the nanocomposites due to the increment in root plastic deformation and shape fractures in certain regions.

Dong et al. [106] worked on the synthesis of a nanocomposite of the thermoplastic resin of polyurethane (CB/TPU) by modifying it with carbon black. Different combinations of nanofiller were loaded into a matrix using a combined co-coagulation method and a hot-pressing technique. The results confirmed the presence of interactive hydrogen bonding between the filler and resin; this resulted in the uniform dispersion of carbon black all through the polymer, yielding improved mechanical properties, thermal stability, and conductivity. The samples were further investigated at impact velocities between 20 and 30 m/s using impact angles of 30° and 120°; the tensile strength and erosion rate were studied. It was observed that maximum and minimum wear rates were recorded at 30° and 90° angles of impact. The results demonstrated a typical ductile erosion performance for the composite material, which indicated that the nanocomposites in question are suitable for use in protective coatings. The homogeneous dispersal of carbon black in the polymer matrix was noted due to the presence of intermolecular hydrogen bonding interactions. This was also associated with a decrease in the glass transition temperature (Tg) from 27.35 °C for neat thermoplastic polyurethane to 30.37 °C for thermoplastic polyurethane containing 12 percent carbon black. The carbon black significantly enhanced the mechanical properties of the neat thermoplastic polyurethane; similarly, the thermal stability and thermal conductivity of the nanocomposites were shown to be enhanced with the increase in carbon black content. The study suggested that the nanocomposites (CB/TPU) are suitable for use as protective coatings and can effectively reduce the occurrences of wear activities in applications where high-speed particles impinge on surfaces.

Suresh et al. [97] studied a polyphenylene sulfide polymer after hybridizing it with short glass fibers as reinforcement. They analyzed its erosion behavior at various weight percentages. Steady-state wear rates were calculated at different erodent velocities (from 25 to 66 m/s) and impact angles (from 15° to 90°) using silica as the erodent (200 ± 50 µm). The morphology of the eroded faces was analyzed using scanning electron microscopy (SEM), and probable wear processes were discussed. In addition, the artificial neural network (ANN) technique was utilized to calculate the wear rate of PPS hybrid composites. The findings show that the theoretical values are very appropriate relative to measured values. The PPS is inherently a brittle thermoplastic polymer, and the erosion values of its nanocomposite improved with an increase in its fiber percentage and the impact velocity of the erodent. The study concluded that the wear property was reliant on the test conditions and the fiber percentage. In addition, the study proved the incredible potential of neural networks for modeling wear/erosion.

Harsha et al. [95] investigated the solid particle erosion behavior of polyetherimide (PEI) composites utilizing randomly distributed short E-glass, carbon fibers, and solid lubricants (PTFE, graphite, and MoS_2_). The erosion rates (ERs) of the PEI composites were calculated at various angles of impingement (from 15° to 90°) and impact velocities (from 30 to 88 m/s). The relationships of the erosion rate with mechanical properties such as tensile strength (T), impact strength (I), ultimate elongation to fracture (e), hardness (H), and shear strength (S) were discussed, and it was proved that these properties influenced the erosion rate of PEI composites. It was demonstrated that neat polyetherimide and its hybrid composites reinforced with carbon fibers displayed semi-ductile behavior, and the highest erosion rate was observed at a 60° impingement angle. However, the PEI composite using glass fiber as its filler reached its maximum erosion rate at a 60° impingement angle under impact velocities of 88 and 30 m/s. These findings demonstrate that fiber content, experimental parameters, and mechanical properties contributed to controlling the erosion rate of the composites. SEM investigation of the eroded surfaces demonstrated that the basis of erosion consisted of the phenomena of microcracking, sand particle embedment, fiber pull-out, chip formation, fiber cracking, and the removal of fibers. It was also concluded that, for the application of PEI in erosive wear environments, it is always desirable to use a lower glass fiber content when reinforcing a neat PEI polymer.

### 3.3. The Role of Filler

For aerospace and engineering applications, the tribological and mechanical benefits of hybrid thermoplastic composites are growing steadily. This is due to their favorable characteristics, including low density, stiffness, and high strength, especially as compared to the common metallic/monolithic metal alloys. The wear resistance of a surface material is an important protection against erosion. Nanotechnology using appropriate filler is the key to designing nanocomposites for high-performance applications. The effects of different types of filler are reported in the literature. Some of the most frequently used fillers are carbon-based because they influence mechanical parameters and lubrication while also improving the erosive performance of the matrix. The tribological and structural usage of glass and carbon fibers has been explored in engineering and aerospace applications; the available information related to erosion is given in Table 2. In some cases, there are polymer composites using glass or carbon fibers that have resulted in poor erosion resistance as compared to neat polymers. Consequently, the wear resistance of such composites is an issue because they exhibited relatively poor resistance compared to metallic substrates. Alternatively, depending upon the nature of the matrix, the mechanical and wear resistance of certain composites was observed to be increased with the addition of a 10 to 30% volume of carbon fiber.

For the most part, nano- and micro-fillers are combined with a matrix to modify and adjust the mechanical parameters, wear challenges, thermal characteristics, and curing procedure of polymers. Nanoparticles can affect the curing process by increasing the initial curing mechanism (autocatalytic reaction); sometimes, in high quantities, they hamper the crosslinking of resin, and curing is inhibited.

Pyrogenic silica is employed in coatings, paints, and adhesives due to its rheology. In particular, nano-silica has a pronounced effect on mechanical characteristics, wear parameters, and cavitation erosion properties, as well as on thermal properties and the curing reaction. Natural fibers have gained enormous attention as reinforcing materials for polymer-based composites. Natural fibers, such as kenaf, coconut, applewood, rice husk, and sisal banana, have been studied, and the effects of using short fiber fillers to reinforce polymers have been explored. Nano- and micro-fillers, such as cork nanoparticles, carbon black, and jute, have been explored for use in epoxy resins to assess their ability to modify the mechanical properties, thermal characteristics, curing reaction, and wear resistance of epoxy resins [91].

Similarly, the addition of CNTs resulted in a considerable enhancement of the modulus properties, tensile strength, fracture resistance, and toughness of the matrix used. Hybrid composites modified with graphite have been reported to yield anti-wear and self-lubrication results compared to neat resin [107,108,109]. Another ideal multipurpose filler is graphene, which has gained significant attention for its production of protective layers for numerous applications of polymer composites; however, the erosion data for these composites were not reported, and there are economic limitations to some of the fillers used.

Chen et al. [98] explored the erosive wear behavior of epoxy composites in the form of films using carbon nanotubes (CNTs). The CNT film composites were synthesized in two different patterns, i.e., unidirectionally (0°) and bi-directionally (0° and 90°) aligned CNTs, and the films were subjected to a particle flow. It was reported that the unidirectional CNT film/epoxy composite exhibited greater erosive wear endurance compared to the bi-directional epoxy composite because of the additional impact energy inclusion caused by the CNT filler network. Wear mechanisms were further examined by scanning electron microscopy (SEM) at different impingement angles. Thus, the work productively established the fabrication of an epoxy–CNT film using traditional composite fabrication processes with minimal erosive wear and high electrical performance, yielding a promising material for engineering applications.

Harsha et al. [41] reported filler compositions from 0 to 30 percent weight and examined the erosion behavior of polyether ketone (PEK) strengthened with short glass fibers and exposed to a silica erodent. The steady-state wear rates were calculated at various impact velocities (from 25 to 66 m/s) and impact angles (from 158° to 908°). It was inferred that neat PEK and its composites with short glass fibers exhibited the highest erosion rate at a 30° impact angle, which is indicative of its ductile erosion performance. It was observed that the erosion performance of PEK composites improved with an increase in the percentage of glass fibers. The erosion rate was also predicted using the artificial neural network technique. The influence of numerous learning algorithms on the neural networks was explored. Utilizing an empirically determined database of PEK composites, a correlation of the predicted erosion rates with the experimentally measured values was observed.

Arjula et al. [110] conducted a solid particle erosion study using polyetherimide (PEI) and a unidirectional carbon-fiber-reinforced PEI (CF/PEI); they employed silica sand particles (200 ± 50 μm) as an erodent at various impact angles (from 15° to 90°) and impact velocities (from 25 to 66 m/s). The impact of the fiber orientation (0° and 90°orientations) on the erosion rate of the CF/PEI composite was also investigated. The neat PEI demonstrated the highest erosion rate at a 30° impact angle, representing ductility under the tested conditions, whereas the CF/PEI composite reached its highest erosion rate at a 60° impact angle and showed semi-ductile behavior. It was concluded from the results that the fiber orientation had a considerable influence on the erosion rate at diagonal impact angles. The erosion rate was greater in the case of a 90°orientation compared to a 0°orientation of fibers. The erosion mechanism was investigated using a scanning electron microscope, and it was demonstrated that surface removal of neat PEI occurred primarily by micro-cutting, microcracking, and plowing at oblique angles, whereases microcracking and plastic deformation were the phenomena that resulted in material loss at a 90° impact angle. PEI is a ductile polymer by nature; however, both brittle and ductile features were observed in the micrographs.

Fouad et al. [111] investigated the solid particle wear behavior and erosive mechanism of epoxy-based unidirectional glass-fiber-modified plastic (GFRP) composites. The wear tests were conducted using irregular silicon carbide (SiC) particles (150 ± 15 m) as an erodent. The wear behavior of the composites was evaluated at three impact angles (30°, 60°, and 90°) under varying conditions of exposure time and pressure. The wear behavior of GFRP showed a conversion from ductile to brittle at a 60° impingement angle, and the wear rate was at its maximum. The eroded surfaces were examined under a scanning electron microscope to study the morphology and evaluate the damage mechanisms.

Abenojar et al. [82] explored the effect of the addition of silica nanoparticles on the wear tolerance of epoxy. It was revealed that the material’s mechanical properties were also affected by the addition of nano-SiO_2_. The composite had lower values for hardness and strength, but its ductility increased. It was concluded that wear was affected by the addition of the nanoparticles. It was suggested that it would be advantageous to use a lower percentage of silica (3%) because the Young’s modulus decreases when a larger content of silica is added, particularly for bulk applications. It was recommended to avoid using silica epoxy nanocomposites for applications that require greater wear resistance.

Suresh et al. [96] evaluated the solid particle erosion performance of a neat polyether ether ketone (PEEK) resin by modifying it with unidirectional glass fibers and carbon fibers. A wear study was performed using impinging silica sand particles (200 ± 50 nm) as an erodent. The steady-state wear rates of the synthesized composites were calculated at different impact velocities and impact angles. It was shown that the neat polymer displayed the highest wear rate at a 30° impingement angle, while its hybrid composites demonstrated semi-ductile performance, reaching their maximum erosion rates at a 60° impingement angle. The carbon-fiber-reinforced composites showed better erosion resistance compared to the glass-fiber-reinforced composites. It was revealed that, at smaller impact angles, fiber orientation had a major impact on the erosion rate. If the erodent struck perpendicular to the fiber orientation, the wear rate of the composites was greater than when parallel to the fibers. Impact velocity was shown to have an important role in governing erosion. Neat polyether ether ketone showed a value of 2.0 for wear rate, whereas its composites yielded magnitudes ranging from 2.4 to 3.0. It was demonstrated that there was erosion anisotropy with fiber orientation at smaller impact angles (15° and 30°). The morphology of the eroded surfaces was investigated under a scanning electron microscope; the highlighted damage mechanisms were the microcracking of fibers, the plastic deformation of the matrix, and fiber–matrix debonding.

Cao et al. [100] studied reduced graphene oxide and synthesized silicone–acrylate resin-reinforced composite films (rGO/SAR) using an in situ synthesis method. The structural characterization of the rGO/SAR composite films was made using the Raman spectrum, an atomic force microscope, scanning electron microscopy, and a thermogravimetric analyzer. The results demonstrated that the rGO was evenly distributed in a silicone–acrylate resin matrix. The bulge test was utilized to demonstrate the effect of rGO loading on the mechanical properties of composite films. Significant improvements in the Young’s modulus and yield stress (290% and 320%, respectively) were observed by combining the rGO with the silicone–acrylate resin. Similarly, the adhesive energy between the composite films and the metal substrate was also enhanced, to around 200%. The rGO loading was also observed to have a large effect on the erosion resistance of the composite films, rendering it suitable mainly for applications in mechanical corrosion coupling environments.

Kumar et al. [112] utilized vinyl ester resin and short E-glass/carbon fibers at various weight fractions (from 20% to 50%) to synthesize fiber-reinforced hybrid composites. The Taguchi orthogonal array design was utilized, and a series of experiments were conducted to investigate the optimal control factors leading to increased erosion efficiency. The steady-state erosion behavior of the composites was determined at different impingement angles, impact velocities, and erodent sizes by keeping other variables constant. The eroded surface of the material was examined by scanning electron microscopy (SEM) to examine the wear mechanisms. The measured mechanical parameters were correlated to the erosion rates of the composites. It was observed that the storage modulus (E0) gradually increased up to 3927 MPa for 40% composition, and on the further addition of fiber, the E0 value decreased to 3321 MPa at 0 °C. The highest storage modulus was observed at 40% fiber content. This behavior was attributed to the maximum stress transfer between the fibers and the matrix at this composition. To estimate the heterogeneity of the system, a Cole–Cole plot of hybrid composites was plotted at various fiber ratios. Finally, a wicket plot was used to examine the behavior of the material in terms of relaxation.

## 4. Drawbacks and Future Needs

Very few comprehensive investigations have used polymers and their hybrid composites for erosion-resistant applications. The data given in Table 1 show that although a detailed study was carried out using thermoset and thermoplastic polymers, some parameters, such as wear coefficient, are still unexamined. Although researchers have reported on composites in thermal, mechanical, and morphological studies relating to the wear of surfaces, gaps are nevertheless observed when calculating the erosion rates. Some of the composites’ reported properties can be related to erosion, as seen in Table 3. Given that there are some common governing parameters for erosion and other applications, the reported materials that have properties related to erosion can also be assessed for wear applications. Only composites using fillers such as glass fibers and carbon fibers have been extensively reported on with respect to erosion variables, as seen in Table 2. However, there is very little research available on estimations of the erosive wear of materials using nanofillers. Moreover, in many research papers, the data presented are derived using only simulation software, and no experimental results are used to verify them. As there are many controlling factors for erosion, in the majority of cases, even if data are available for certain composite materials, the calculated parameters differ for other composites of interest; therefore, there is still a need to explore more in this field and to obtain harmonized and adequate data. In particular, in this area of study, new combinations of materials should be explored.

## 5. Conclusions

Polymers and their composites have been utilized as structural and tribo-materials for erosion protection purposes in numerous industries owing to their advantageous characteristics, such as high strength, greater hardness, enhanced toughness, high modulus, and, in some cases, recyclability. The most beneficial features of polymers are their flexibility in design/shape and the ease of manufacturing them into intricate parts by extrusion or injection molding techniques. In the context of many fundamental applications, erosion is a major concern because it causes the failure of parts and components. The present work is a review of thermoplastic and thermoset polymeric composites that have been explored in the previous few years for their erosion/wear-related qualities. The polymer composites of EP, PU, PEI, PES, PEEK, PEK, PS, and BMI have been discussed, along with the particulars of solid particle fillers, such as clay alumina, titanium oxide, zinc oxide, silica, and carbon-based fillers. Based on the findings relating to the impacts of the mentioned fillers on the characteristics of composite materials with respect to solid particle erosive wear, the key conclusions of this work can be listed as follows:Epoxy is a versatile coating with certain disadvantages, including brittleness and rigidity, that can be efficiently reduced by using hybrid polymers, i.e., its modification with polyurethane for surface protection.Nano–Al_2_O_3_ has the ability to benefit from hydrogen bonding with polymer chains; because of its low cost, it can be considered a promising nanoparticle for increasing wear resistivity.Carbon-based materials, such as MWCNT, graphene, and graphite, can have covalent bonding within their matrix; thus, they can improve the modulus and tensile strength of the polymer matrix. However, graphite can exhibit Van der Walls bonding among its layers, so it may reduce the shear stress of the composite. Comparing graphene with MWCNT, we observe its better dispersion in the matrix in comparison with MWCNT.Replacing carbon fiber with aligned CNT films as the reinforcement material can lead to a change in erosive wear behavior from brittle to ductile. CNTs can absorb more energy upon fracture or bending, and it is possible to further functionalize CNTs to increase their erosive wear.The erosive wear of composites with glass fibers as a filler is greater than that of carbon fiber composites. The change in erosive rate is due to fiber/matrix interfacial bonding and the different properties of the fibers.Due to its minimal cost, nano-silica is recognized as the most widely used NP for manufacturing erosive endurance composites. It can develop effective bonding with the matrix because of its self-lubricating characteristics.Polymer hybrids containing nano–TiO_2_ can alter mechanical characteristics and erosive properties by effectively transmitting the cracks and developing a strong bridge between the filler and the polymer resin during the erosion process.ZnO nanoparticles have the major advantage of favorable mechanical properties in polymer hybrid materials; with a phase disruption mechanism, its surface functionalization can develop strong covalent bonds between polymer chains.The addition of nano-clay to a polymer matrix results in a phase disruption mechanism that can enhance the mechanical characteristics of the polymer composite to some extent by utilizing its Van der Wall interactions.For wear applications, the filler plays a key role in optimizing the polymer; important factors include its composition, the polymer–filler interaction, and the homogeneous distribution of filler in the matrix. The erosion of the matrix surface varies depending on the nature of the erodent and the impingement conditions.At perpendicular angles, erosion was found to be lower; at inclined ranges, it varied due to the slicing of fiber material.Higher hardness values of composites do not indicate a higher wear resistance value. Lower Tg values suggest that the nanocomposites plasticize more effectively with the neat resin and convey better erosion resistance. The observed plasticization is in harmony with the greater strength and augmented ductility.

## Figures and Tables

**Figure 1 nanomaterials-12-02194-f001:**
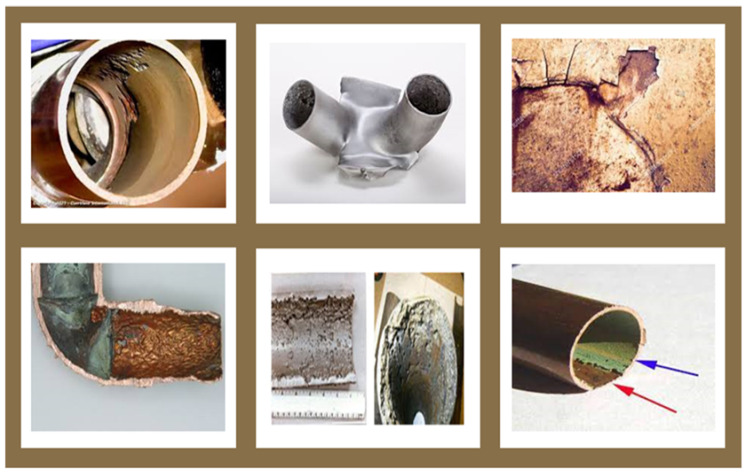
Erosion effects on surfaces.

**Figure 2 nanomaterials-12-02194-f002:**
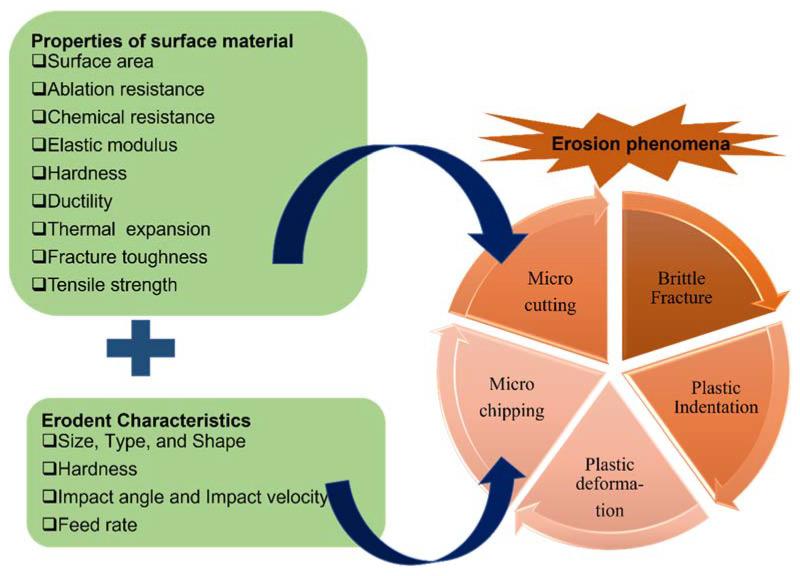
Factors governing erosion phenomena.

**Figure 3 nanomaterials-12-02194-f003:**
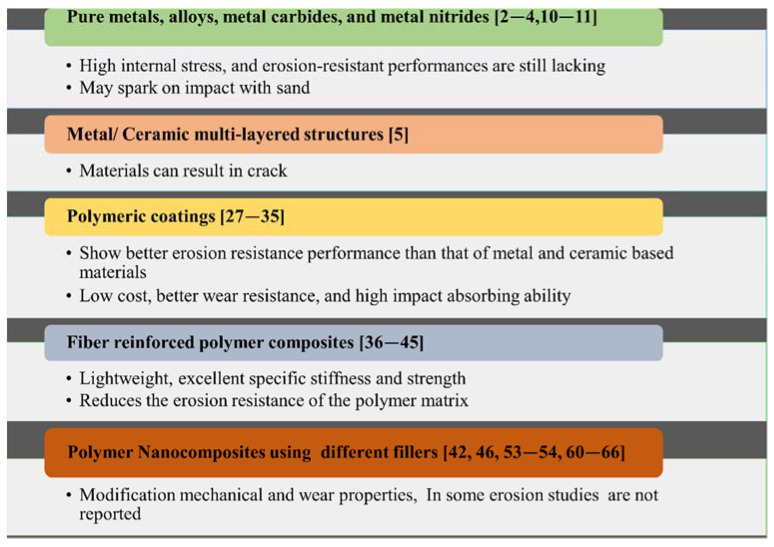
Historical utilization of materials for surface protection.

**Figure 4 nanomaterials-12-02194-f004:**
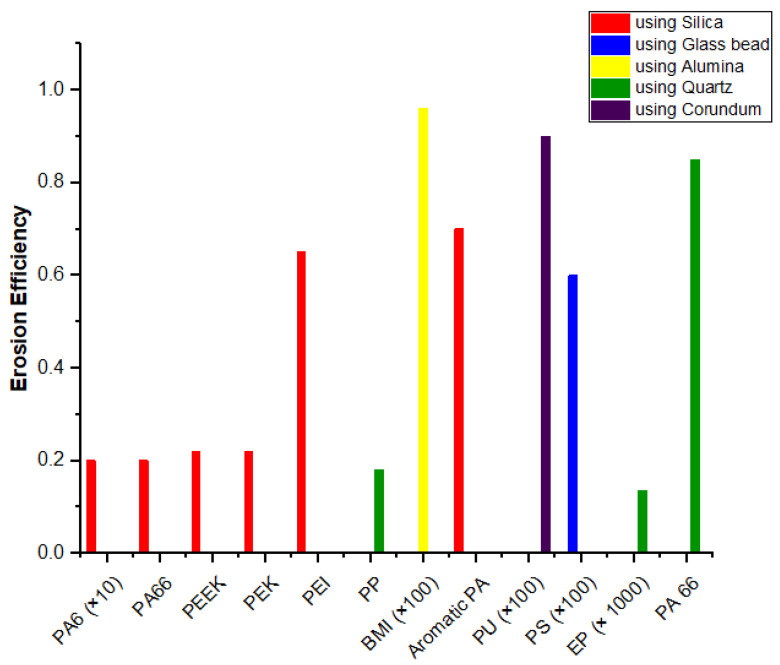
Minimum reported erosion efficiencies of neat polymers at optimized impact velocity using different erodents [15,24].

## Data Availability

All data are available within the manuscript.

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
