# Peer review of "Anti-Wear and Anti-Erosive Properties of Polymers and Their Hybrid Composites: A Critical Review of Findings and Needs"

_nanomaterials, 2022, doi:10.3390/nano12132194_

Round 1
Reviewer 1 Report
I have read the revised version of the Manuscript and found that the authors have taken into accounts the concerns that I raised. Thus I recommend it for publication.
Author Response
Thank you for the recommendation of the manuscript.
Reviewer 2 Report
The review is focused on the development of the wear properties of polymers and their hybrid composites. The gaps for the identification of their effective use in surfaces exposed to wear environments was also approached.
I already reviewed this manuscript twice, it is now the third time and I realize that most of comments were not correctly addressed. Therefore, I reinforce the major comments that must be addressed.
- The aim of the review should be clearly defined as well as the timeframe, years and the readers target (senior or junior reasearchers? Industry?). This information should be indicated in the introduction and abstract.
- Without defining a timeframe it seems that the review does not reflect the actual state of the art since it is not focused.
- Generally the manuscript needs to be focused and reviewed in order to provide a critical analysis instead of listing some publications. The assessment of the information indicated in the tables 1, 2 and 3 must be discussed. Are the most relevant? Are within a certain timeframe? Are the best results reported? The tables aren't even cited in the text manuscript...
Author Response
Please find the file enclosed.

Reviewer 3 Report
The changes made to the text in different colours make it impossible to read and understand.
The text of this review is very hard to follow, and the order of the chapters is illogical. First, you talk about polymer coatings, then you mention that other materials were used historically, then you go back to the polymers, but never once is there any mention of what type of polymers are used.
Some figures have numbers but some do not, why? Figure 3 is copied in twice in various sizes.
Please, clean up the manuscript, prepare it properly for review and then resubmit, this is a waste of my time.
Author Response
Please find the file enclosed.

Reviewer 4 Report
1The paper is enough good review of the present state in the polymer composite erosive properties. Before publication, some points should be revised:
11) Section 1.2. The sentence “To achieve these parameters synthesizing nanocomposites utilizing erosion-resistant polymers i.e., thermoplastic/ ductile polymers is the finest choice” should be justified. Why namely NANOcomposites? Why namely thermoplastic/ ductile polymer? Please describe it here.
22) Line 144. The dot before bracket should be removed.
33) Line 244. The uppercase letter is not needed
44) Lines 354-364. The nature of nanofillers advantage in relation to coarse powder fillers should be described and justified in detail.
55) Line 404. The uppercase letter is not needed
66) Line 476. The comma should be removed
77) Line 499. It’s better not to use the term “Nanotechnology” in this case
88) Line 556. The uppercase letter is not needed
Author Response
Please find the file enclosed.

Round 2
Reviewer 2 Report
All the issues raised were addressed.
This manuscript is a resubmission of an earlier submission. The following is a list of the peer review reports and author responses from that submission.
Round 1
Reviewer 1 Report
The review is focused on the development of the wear properties of polymers and their hybrid composites. The gaps for the identification of their effective use in surfaces exposed to wear environments was also approached. The manuscript is within the scope of Nanomaterials, however, before publication some major concerns must be addressed, namely:
- The aim of the review should be clearly defined as well as the timeframe and the readers target (senior or junior reasearchers? Industry?). This information should be indicated in the introduction and not in a different section.
- Without defining a timeframe and only stating a "few years" it seems that the review does not reflect the actual state of the art since it is not focused.
- Generally the manuscript needs to be focused and reviewed in order to provide a critical analysis instead of listing some publications. The reason behind tha information indicated in the tables 4.1, 4.2 and 4.3. Are the most relevant? Are within a certain timeframe? Are the best results reported? The year of such publications should be include in the tables.
Author Response
Kindly find the enclosed file against the suggestions.

Reviewer 2 Report
This work summarized the anti-wear/erosive properties of polymers and their hybrid composites: A critical review on findings and needs. This manuscript can be published but needs to be revised as follows:
- The language in the manuscript needs to be revised accordingly.
- The Abstract part needs to write the innovation of the work.
- The choice of keywords is not accurate enough.
- The clarity of the pictures in the manuscript is low.
- The conclusion should be written in points.
- The font size format of part 4.1 needs to be unified.
- It is suggested to refer some relevant latest literatures: Colloids and Surfaces A: Physicochemical and Engineering Aspects 626 (2021) 127066; Materials Today Communications 22 (2020) 100800.
Author Response

(The authors gave the same response as above.)

Author Response

(The authors gave the same response as above.)

Round 2
Reviewer 1 Report
My comments were not well addressed. Therefore, I do not support publication in the present form. It does not make any sense have a section separated from the introduction just to mention the aim of the review. Moreover, this is not focused nor limited to a timeframe.
Author Response
Kindly find the attached file.

Reviewer 2 Report
I have read the revised version of the Manuscript and found that the authors have taken into accounts the concerns that I raised. Thus I recommend it for publication.
Author Response
Kindly find the file attached.
